# Three-Dimensional Angiographic Characteristics versus Functional Stenosis Severity in Fractional and Coronary Flow Reserve Discordance: A DEFINE FLOW Sub Study

**DOI:** 10.3390/diagnostics12071770

**Published:** 2022-07-21

**Authors:** Valerie Stegehuis, Jelmer Westra, Coen Boerhout, Martin Sejr-Hansen, Ashkan Eftekhari, Hernan Mejía-Renteria, Maribel Cambero-Madera, Niels Van Royen, Hitoshi Matsuo, Masafumi Nakayama, Maria Siebes, Evald Høj Christiansen, Tim Van de Hoef, Jan Piek

**Affiliations:** 1Amsterdam UMC—Location AMC, Department of Cardiology, University of Amsterdam, Heart Center, Amsterdam Cardiovascular Sciences, 1105 AZ Amsterdam, The Netherlands; v.e.stegehuis@amsterdamumc.n (V.S.); c.k.boerhout@amsterdamumc.nl (C.B.); t.p.vandehoef@amsterdamumc.nl (T.V.d.H.); 2Department of Cardiology, Aarhus University Hospital, Skejby, 8200 Aarhus, Denmark; jelmer.westra@clin.au.dk (J.W.); sejr@clin.au.dk (M.S.-H.); asef@rn.dk (A.E.); evald.christiansen@dadlnet.dk (E.C.); 3Department of Cardiology, Hospital Clínico San Carlos IDISSC and Universidad Complutense de Madrid, 28040 Madrid, Spain; hernan_m_r@yahoo.com; 4Tergooi Hospital, 1261 AN Blaricum, The Netherlands; mmaderacambero@tergooi.nl; 5Department of Cardiology, Radboud University Nijmegen, 6525 XZ Nijmegen, The Netherlands; niels.vanroyen@radboudumc.nl; 6Gifu Heart Center, Gifu 500-8384, Japan; matsuo@heart-center.or.jp (H.M.); masafumi331@gmail.com (M.N.); 7Japan Toda Chuo General Hospital, Toda 335-0023, Japan; 8Amsterdam UMC—Location AMC, Department of Biomedical Engineering and Physics, 1105 AZ Amsterdam, The Netherlands; m.siebes@amsterdamumc.nl

**Keywords:** coronary, physiology, QCA, fractional flow reserve, coronary flow reserve

## Abstract

Background: Coronary angiography alone is insufficient to identify lesions associated with myocardial ischemia that may benefit from revascularization. Coronary physiology parameters may improve clinical decision making in addition to coronary angiography, but the association between 2D and 3D qualitative coronary angiography (QCA) and invasive pressure and flow measurements is yet to be elucidated. Methods: We associated invasive fractional flow reserve (FFR), coronary flow reserve (CFR) and coronary flow capacity (CFC) with 2D- and 3D-QCA in 430 intermediate lesions of 366 patients. Results: Overall, 2D-QCA analysis resulted in less severe stenosis severity compared with 3D-QCA analysis. FFR+/CFR− lesions had similar 3D-QCA characteristics as FFR+/CFR+ lesions. In contrast, vessels with FFR−/CFR+ discordance had 3D-QCA characteristics similar to those of vessels with concordant FFR−/CFR−. Contrarily, FFR+/CFR− lesions had CFC similar to that of as FFR-/CFR- lesions. Conclusions: Non-flow-limiting lesions (FFR+/CFR−) have 3D-QCA characteristics similar to those of FFR+/CFR+, but the majority are not associated with inducible myocardial ischemia as determined by invasive CFC. FFR−/CFR+ lesions have 3D-QCA characteristics similar to those of FFR−/CFR− lesions but are more frequently associated with a moderately to severely reduced CFC, illustrating the angiographic–functional mismatch in discordant lesions.

## 1. Introduction

Coronary angiography (CAG) has been well-documented to provide limited accuracy in identifying hemodynamically significant coronary stenosis [1,2,3,4,5]. For this purpose, contemporary clinical practice guidelines recommend invasive physiology assessment to identify functional stenosis severity using fractional flow reserve (FFR) or instantaneous wave-free ratio (iFR) when non-invasive evidence of ischemia is not available [6]. However, invasive assessment of FFR and iFR requires instrumentation of the coronary artery with sensor-equipped guide wires and is, therefore, more time-consuming and costly than sole angiographic assessment. Consequently, decision-making on coronary revascularization remains dominantly guided by visual estimation of stenosis severity by coronary angiography. Despite the well-documented mismatch between visual angiographic stenosis appearance and FFR values [7], two aspects deserve further evaluation. First, contemporary quantitative angiography analysis (QCA) allows 3-dimensional vessel reconstruction to provide more detail regarding anatomical stenosis severity, which may provide a better estimation of functional stenosis severity [8]. Second, FFR is known to be influenced by microvascular resistance, where, for a given stenosis, FFR increases with increasing microvascular resistance [9,10]. Therefore, a comparison with FFR alone may lead to erroneous conclusions regarding the ability of angiography to define functional stenosis severity. The combined assessment of coronary pressure and flow allows calculation of both FFR and coronary flow velocity reserve (CFR), which together provide more insight into the functional relevance of coronary artery disease. Moreover, these measurements allow the calculation of stenosis resistance indices and coronary flow capacity as robust markers of functional stenosis severity and vessel flow characteristics, respectively. Therefore, we sought to define the association of 3D-QCA angiographic stenosis characteristics and comprehensive invasive physiology using FFR/CFR agreement, as well as stenosis resistance and CFC.

## 2. Methods

### 2.1. Patient Population

Between June 2015 and November 2017, 456 patients were enrolled in the DEFINE-FLOW (Distal Evaluation of Functional performance with Intravascular sensors to assess the Narrowing Effect–combined pressure and Doppler Flow velocity measurements) study. For this sub analysis, 430 intermediate lesions of 366 patients with measurements with sufficient quality, as evaluated by an independent core lab, were analysed. This prospective, non-blinded, non-randomized, multi-centre trial enrolled subjects with at least one coronary lesion undergoing physiologic evaluation as per routine clinical practice. Rationale and design have been published elsewhere [11]. Briefly, subjects had to be 18 years or older at time of inclusion, with at least one epicardial stenosis of ≥50% diameter (by visual or quantitative assessment), in a native coronary artery, ≥2.5 mm reference diameter and supplying sufficiently viable myocardium. Important exclusion criteria were recent (within 3 weeks prior to cardiac catheterization) ST-segment elevation myocardial infarction (STEMI), a chronic total occlusion, prior coronary artery bypass grafting (CABG), left main coronary artery disease requiring revascularization, extremely tortuous or calcified coronary arteries, known severe left ventricle hypertrophy and planned need for cardiac surgery. A total of 12 centres in Europe and Japan with ample experience as proven by previously conducted Doppler flow studies in intracoronary Doppler flow velocity measurements recruited subjects. The trial was conducted in accordance with the Declaration of Helsinki and all applicable local regulations. Every subject gave written informed consent prior to enrolment. 

### 2.2. Cardiac Catheterization and Physiological Assessment

Intracoronary 100 to 300 µg nitroglycerin was administered at the beginning of the procedure, repeated every 30 min if indicated, to avoid catheter or wire induced coronary spasm and minimize flow-mediated dilation. After diagnostic angiography, a 0.014′′ dual pressure and Doppler flow velocity sensor guidewire (ComboWire XT; Philips Volcano, San Diego, CA, USA) was zeroed at atmospheric pressure and calibrated to aortic pressure at the ostium of the guiding catheter. Subsequently, the guidewire was positioned at least five vessel diameters distal to the lesion. Before inducing hyperemia, a stable flow signal was obtained and the position of the guidewire was documented fluoroscopically. Hyperemia was induced by an intracoronary bolus of 100 µg adenosine for both left and right coronary arteries, followed by a flush of saline [12,13]. Measurements were repeated at least once using the same dose of adenosine. An independent corelab assessed the quality of pressure and flow, and solely corelab approved measurements were used in the present sub study. At the end of the procedure, the guidewire was returned to the guiding catheter to assess pressure drift, where Pd/Pa ± 0.02 triggered re-assessment. FFR was calculated as the mean distal (Pd) to aortic pressure (Pa) during hyperemia, and CFR was calculated as the ratio of hyperemic average peak flow velocity (APV) by baseline APV. FFR ≤ 0.8 and CFR < 2.0 were considered abnormal (Table 1). Normal CFC was defined as a CFR ≥ 2.8 and hAPV of ≥49.0 cm/s. Mildly reduced CFC was defined as a CFR < 2.8 but > 2.1 and hAPV of <49.0 but >33.0 cm/s. Moderately reduced CFC was defined as CFR ≤ 2.1 and > 1.7, and hAPV ≤ 33.0 and > 26.0 cm/s. Finally, severely reduced CFC was defined as a CFR ≤ 1.7, and hAPV ≤ 26.0 cm/s [14]. Normal to mildly reduced CFC was considered normal, whereas moderately and severely reduced CFC were considered abnormal. 

### 2.3. Quantitative Coronary Angiography

Three-dimensional quantitative coronary angiography (3D-QCA) was performed by three blinded physicians (JW, MSH and HMR) at the Interventional Coronary Imaging Core Laboratory; Aarhus University Hospital, Skejby, Aarhus, Denmark and Hospital Clinico San Carlos, Madrid, Spain using QAngio XA (Medis Medical Imaging Systems bv., Leiden, The Netherlands). If 3D-QCA was not feasible, 2D-QCA was used. 3D-QCA analysis was performed based on automated calibration using two images ≥ 25° separated. 2D-QCA was performed based on catheter calibration using the angiographic view with best exposure of the lesion severity. The same algorithms were used for vessel edge detection. Lesions were stratified according to the SYNTAX classification: right coronary artery (RCA–segments 1, 2, 3), left anterior descending artery (LAD–segments 6, 7 and 8) and the left circumflex artery (LCX–segments 11, 12 and 13). Angiographic diameter stenosis (DS), minimal lumen diameter (MLD), area stenosis (AS), lesion length and reference diameter were reported. 

### 2.4. Statistical Analysis

All analyses were performed on the lesion level, except for baseline patient characteristics. Normality of the data was tested using the Shapiro–Wilk test. Continuous variables are presented as median (Q1, Q3) or frequency (percentage), where appropriate. Categorical variables are presented as proportions and were analysed with a chi-square test. Angiographic lesion severity per category and the respective FFR and CFR value of each specific lesion were plotted in a box-and-whisker plot to show the degree of dispersion and skewness in the data and to identify potential outliers. The box-and-whisker plots were created with GraphPad Prism version 8.3.0 (GraphPad Software Inc., La Jolla, CA, USA). A Kruskal–Wallis test with pairwise post hoc Bonferroni correction for multiple comparisons was performed to identify baseline and QCA differences between groups. QCA-derived DS was compared to per-procedural visual estimated DS. An independent samples T-test was used to compare 2D- with 3D-QCA analysis within the four groups based on FFR and CFR. A *p* value < 0.05 was considered statistically significant. All statistical analyses were performed in STATA version 15.1 (StataCorp, College Station, TX, USA).

## 3. Results

### 3.1. Patient and Lesion Characteristics

Baseline patient characteristics (n = 366) are shown in Table 2. The mean age was 67 ± 10.1 years, 74% were male, and the majority of interrogated vessels comprised the LAD (66%). Overall mean FFR was 0.82 ± 0.1, and overall mean CFR was 2.3 ± 0.7. There was no significant difference in proximal, mid or distal lesions between FFR/CFR groups (*p* > 0.05). Patients with FFR+/CFR− were significantly younger compared with patients with FFR−/CFR+: mean age 64.8 ± 10.6 versus 68.8 ± 10.8 years (*p* = 0.021).

### 3.2. Percentage Diameter Stenosis by Visual Estimation

For a total of 381 lesions, angiographic visually estimated DS was available: 62% (n = 238) were categorized 40–70% DS, 30% (n = 113) were categorized 70–90% DS, and 8% (n = 30) were categorized ≥ 90% DS. Figure 1 shows the dispersion of FFR and CFR values in the three DS categories as visually estimated during CAG. 

### 3.3. Comparison of 2D- versus 3D-QCA Analysis

Table 3 details clinical and angiographic characteristics for all four FFR/CFR groups. Overall, 2D-QCA resulted in lower stenosis severity as determined by DS%, lesion length, MLD and mean RLD compared with 3D-QCA. FFR+/CFR+ lesions were more severely narrowed as determined by DS%, MLD and area stenosis compared with the other groups, both for 2D- and 3D-QCA analysis (mean 3D QCA-DS 54 ± 7.3% versus 59 ± 9.5%, median MLD 0.8 mm (0.6, 1.1) and median area stenosis 82% (70.8, 85.3)). Lesion length was not different between groups for both 2D-QCA and 3D-QCA analysis (*p* = 0.27, and *p* = 0.30, respectively), but within each group lesion length was significantly longer for 3D-QCA versus 2D-QCA analysis (*p* < 0.005 for all FFR/CFR groups). 

### 3.4. 3D-QCA versus Functional Stenosis Characteristics across FFR/CFR Groups

3D-QCA analysis revealed that vessels with FFR+/CFR− discordance had 3D-QCA characteristics similar to those of vessels with concordant FFR+/CFR+: mean 3D-QCA DS% 61 ± 11.1% versus 53 ± 11.6% and median 3D-QCA AS% 73.3% (60.6, 81.2) versus 82% (70.8, 85.3) (*p* > 0.05 for all), with the exception of 3D-QCA MLD (0.8 (0.6, 1.1) versus 1.2 mm (0.9, 1.5)) (*p* < 0.05). Vessels with FFR−/CFR+ discordance had 3D-QCA characteristics similar to those of vessels with concordant FFR−/CFR−: mean 3D-QCA DS% 44 ± 10.3% versus 43 ± 13.1%, median 3D-QCA MLD 1.4 mm (1.2, 1.7) and 1.5 mm (1.2, 1.8) and median 3D-QCA AS% 62.9% (53, 71.3) versus 61.9% (50.3, 72.3) (*p* > 0.05 for all) (Table 2). Of note, FFR+/CFR− had the longest 3D-QCA lesion length (*p* < 0.001): median 23.4 mm (13.3, 29.6), similar to FFR+/CFR+ lesions: median 17.9 mm (13.7, 26.7), whereas FFR−/CFR+ lesions had the shortest lesion length, similar to FFR−/CFR− lesions: median 13.4 mm (9.2, 20.6) versus 16.3 mm (10.4, 23) (*p* > 0.05 for all). Overall, only FFR+/CFR+ lesions had significantly lower 3D-QCA mean RLD compared with FFR−/CFR− lesions (*p* = 0.001). Compared with the other groups, 3D-QCA mean RLD was higher for FFR−/CFR+ and FFR−/CFR− lesions (median 2.7 mm (2.3, 3) versus 2.7 mm (2.4, 3.2), *p* = 0.081) compared with FFR+/CFR+ and FFR+/CFR+ lesions (median 2.5 mm (2.3, 2.7) versus 2.3 mm (2.1, 2.6), *p* = 0.092).

### 3.5. Association of FFR, CFR and CFC

Figure 2 shows the distribution of CFC categories across the FFR/CFR groups. FFR+/CFR− lesions had normal to mildly reduced CFC similar to that of FFR−/CFR− lesions: 95% (n = 77) versus 94% (n = 202) (*p* > 0.05). Despite the lower stenosis severity as assessed by 3D-QCA, FFR−/CFR+ lesions more frequently were associated with a moderately to severely reduced CFC compared with FFR+/CFR− lesions: 56% (n = 34) versus 5% (n = 4) (*p* < 0.001). The majority of FFR+/CFR+ lesions (77%, n = 55) had a moderately to severely reduced CFC, but still 23% (n = 16) of lesions were associated with a normal to mildly reduced CFC. 

## 4. Discussion

This prospective cohort study is one of the first assessing the difference between 2D- and 3D-QCA analysis in angiographic intermediate lesions with combined pressure and flow measurements. Non-flow-limiting lesions (FFR+/CFR−) had 3D-QCA characteristics similar to those of FFR+/CFR+, but the majority were not associated with inducible myocardial ischemia as determined by invasive CFC. FFR−/CFR+ lesions had 3D-QCA characteristics similar to those of FFR−/CFR− lesions, but were more frequently associated with a moderately to severely reduced CFC, illustrating the angiographic–functional mismatch in discordant lesions.

### 4.1. Association of 3D-QCA and FFR/CFR Discordance

The visual–functional mismatch between coronary angiography and FFR has been acknowledged in previous studies, showing a large dispersion of FFR values, especially in angiographic intermediate lesions [15,16,17]. This was confirmed by the present study for both FFR and CFR, as there was a poor correlation between visual DS and invasively assessed FFR and CFR values (Figure 1). Despite the limitations of coronary angiography in assessing lesion severity [18], angiography still remains the cornerstone of decision-making in revascularization, as adoption of coronary physiology parameters in clinical practice is low [19,20]. 

Aside from lesion-specific characteristics such as QCA DS%, MLD and mean RLD, several clinical factors, such as age and sex, and physiological factors are known to contribute to the visual–functional mismatch [17,21,22]. Yonetsu et al. assessed the visual–functional mismatch between 2D-QCA, FFR, CFR and the index of microcirculatory resistance in 849 lesions of 532 patients and found LAD lesions, greater QCA DS%, lower QCA mean RLD, lower CFR and lower index of microcirculatory resistance as predictors of mismatch (DS > 50% and FFR > 0.80), and lower QCA-DS, shorter QCA lesion length, greater QCA mean RLD, higher CFR and higher index of microcirculatory resistance as predictors of reverse mismatch (DS ≤ 50% and FFR ≤ 0.80). In the present study, the main QCA-derived characteristics of FFR+/CFR− or non-flow-limiting lesions were higher 3D-QCA DS%, greater MLD and longer lesion length versus FFR−/CFR+ or “flow limiting” lesions (FFR+/CFR+), which were associated with a lower 3D-QCA DS%, lower AS and shorter lesion length. We did not find a significant difference in the number of LAD lesions across the FFR/CFR groups, although other studies report the highest percentage of mismatch in LAD lesions, potentially since LAD lesions supply the largest myocardial perfusion area compared with the other vessels [23]. Nonetheless, the study by Yonetsu et al. identified the impact of CFR both on mismatch and reverse mismatch. The present study confirms but also strengthens the important role of CFR in the visual–functional mismatch between FFR, angiography and inducible myocardial ischemia: despite the 3D-QCA characteristics associated with greater stenosis severity, non-flow-limiting FFR+/CFR− lesions in the present study had preserved coronary flow as determined by CFR values above the clinical cut-off value of ≥2.0. Since coronary flow is a critical determinant of myocardial ischemia [24], non-flow-limiting lesions have a benign long-term prognosis, whereas flow-limiting lesions are associated with a higher risk of MACE [25]. Although the benefit of FFR-guided PCI over angiography-guided PCI has been established for alleviation of angina and improvement in quality of life [26], non-flow-limiting FFR+/CFR− lesions in the present study were not associated with flow abnormalities despite their angiographic severity. This may explain why in large FFR-guided clinical studies, such as the DEFER and FAME studies, a significant proportion of patients with FFR > 0.80 required revascularization within 5 years of follow-up, and 70% of patients with FFR ≤ 0.80 did suffer from MACE and 50% required repeated revascularization [27,28]. These findings suggest that combined pressure and flow measurement can more accurately identify lesions associated with impaired coronary flow compared with angiography and pressure assessment alone. Novel developments in stenosis severity assessment, such as quantitative flow ratio (QFR) using FFR as the reference standard [29], should be evaluated in this light to accurately identify their diagnostic characteristics versus comprehensive physiological assessment.

### 4.2. Association of CFC with FFR/CFR Discordance

CFC constitutes a relatively novel physiological parameter based on hyperemic flow and CFR, which has important diagnostic and prognostic value [14,30,31]. Moreover, CFC was found to be a better predictor of MACE than CFR, and the risk of MACE decreases significantly after revascularization of lesions associated with a severely reduced CFC [31]. Despite the similar 3D-QCA angiographic characteristics in this study of non-flow-limiting FFR+/CFR− and FFR+/CFR− lesions, invasive flow assessment revealed that the majority of non-flow-limiting lesions have a normal to mildly reduced CFC comparable with FFR−/CFR− lesions. Hence, the majority of FFR+/CFR− lesions may be deemed severe by angiographic parameters, but are not associated with abnormal coronary flow characteristics as determined by CFC [32].

Contrarily, lesions with FFR−/CFR+ show CFC with a higher incidence of moderate to severe CFC that is line with earlier observations that these lesions are more likely to benefit from intervention. Finally, the highest incidence of moderate to severe CFC occurs in FFR+/CFR+ lesions, illustrating the usefulness of CFC for clinical decision making.

### 4.3. Clinical Implications

Although the poor correlation between quantitative angiography and invasive coronary physiology has been well-documented [33,34], the clinical adoption of coronary physiological parameters worldwide is still relatively low. The present study shows that there are differences between 2D-QCA and 3D-QCA. Coronary lesion severity is less by 2D-QCA analysis as compared to 3D-QCA analysis. The lowest MLD by 3D-QCA analysis is in the group with FFR+/CFR+, which may indicate that MLD can be used as a parameter of ischemia as determined by CFC. However, Figure 2 illustrates that the MLD in the other lesion groups does not reflect functional lesion severity as determined by CFC. For example, the majority of FFR+/CFR− lesions, angiographically apparently severe lesions by 3D-QCA, are associated with a normal or mildly reduced CFC that illustrates the angiographic and functional mismatch, also by using 3D-QCA analysis. Coronary physiology measurements, and CFR in particular, have important additional value in guiding clinical decision making in addition to quantitative coronary angiography alone.

### 4.4. Limitations

First, this prospective non-randomized cohort study was not powered for the present analysis, but is the largest analysis of the visual–functional mismatch in FFR and CFR to date. Randomized clinical trials are needed to confirm the findings of the present study. Second, solely corelab approved measurements with sufficient pressure and flow quality signals were included in this analysis, since reliable FFR and CFR values were important for correct classification in the different groups.

## 5. Conclusions

Non-flow-limiting lesions with FFR+/CFR− have 3D-QCA characteristics similar to those of FFR+/CFR+ lesions, but the majority are not associated with reversible ischemia as determined by invasive CFC. Contrarily, FFR−/CFR+ lesions have 3D-QCA characteristics similar to those of FFR−/CFR− lesions, but are more frequently associated with reversible ischemia as determined by CFC.

## Figures and Tables

**Figure 1 diagnostics-12-01770-f001:**
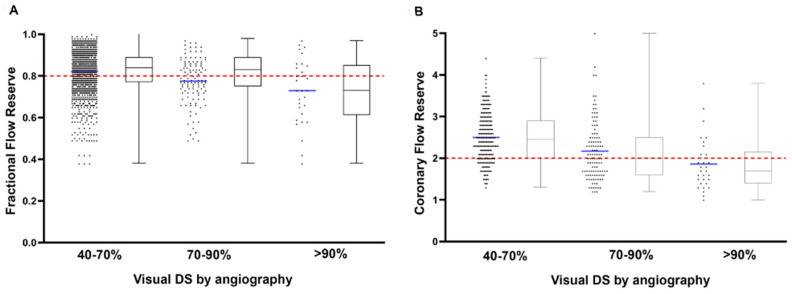
Box-and-whisker and scatter plots showing the FFR (**A**) and CFR (**B**) values of all lesions in the category of DS 50–70%, DS 70–90% and DS > 90% as determined by visual estimation during coronary angiography. The red dashed horizontal lines represent clinical cut-off values of FFR (≤0.80) and CFR (<2.0).

**Figure 2 diagnostics-12-01770-f002:**
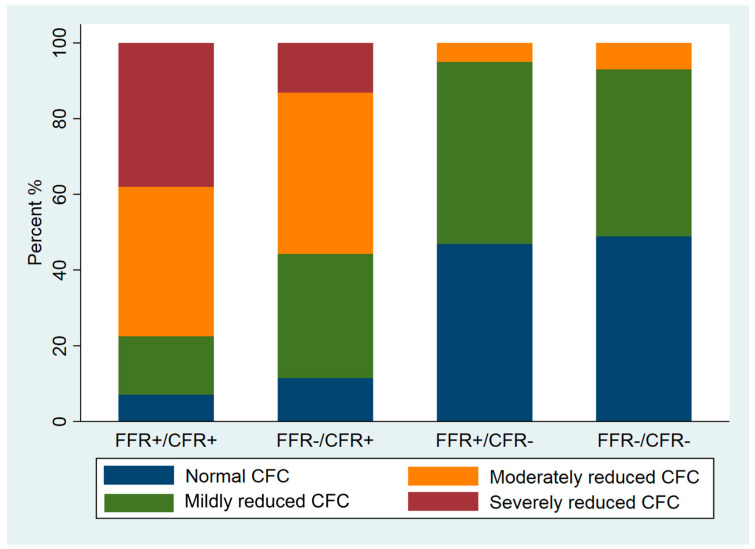
Stack bar graph of CFC frequency in all four FFR/CFR groups.

**Table 1 diagnostics-12-01770-t001:** Abbreviations of different subgroups determined by FFR and CFR.

**FFR +**	**FFR ≤ 0.80**
**FFR −**	FFR > 0.80
**CFR +**	CFR < 2.0
**CFR −**	CFR ≥ 2.0

**Table 2 diagnostics-12-01770-t002:** Baseline clinical characteristics.

Clinical characteristics (n = 366)	
Age, years	66.8 ± 10.1
Male	304 (74%)
LVEF, %	60 ± 8
Hypertension	283 (68%)
Smoking (current)	136 (34%)
Dyslipidemia	375 (88%)
Renal disease	36 (8%)
Diabetes mellitus	115 (27%)
Previous PCI	174 (41%)
Previous MI	116 (27%)
Family history	150 (38%)
Lesion characteristics (n = 430)	
Left anterior descending artery (LAD)	283 (66%)
Left circumflex artery	80 (19%)
Right coronary artery	65 (15%)
FFR	0.84 [0.76, 0.89]
FFR ≤ 0.80	0.73 [0.67, 0.77]
CFR	2.3 [1.9, 2.8]
CFR < 2.0	1.7 [1.5, 1.8]

Values are expressed as mean ± SD, median (Q1, Q3) or n (%).

**Table 3 diagnostics-12-01770-t003:** Clinical and angiographic parameters in 430 intermediate coronary lesions.

Lesions	FFR+/CFR+	FFR+/CFR−	FFR−/CFR+	FFR−/CFR−	*p*-Value across Groups
Patient characteristics	N = 63	N = 71	N = 47	N = 185	
Age, years	66.7 ± 9.9	64.8 ± 10.6	68.8 ± 10.8	66.6 ± 9.9	0.112
Male, n (%)	48 (76%)	55 (81%)	35 (80%)	127 (72%)	0.705
Hypertension	40 (65%)	47 (67%)	36 (77%)	119 (66%)	0.752
Smoking (current)	24 (41%)	23 (34%)	13 (29%)	58 (34%)	0.107
Dyslipidemia	57 (92%)	68 (96%)	42 (89%)	158 (86%)	0.548
Renal disease	4 (6%)	6 (8%)	5 (11%)	17 (9%)	0.946
Diabetes mellitus	26 (41%)	17 (24%)	12 (26%)	46 (25%)	0.301
Previous PCI	28 (44%)	28 (39%)	20 (43%)	68 (37%)	0.722
Previous MI	20 (32%)	20 (28%)	14 (30%)	41 (22%)	0.688
Family history	19 (33%)	30 (44%)	18 (43%)	65 (37%)	0.801
Lesion characteristics	N = 71	N = 81	N = 61	N = 217	
Median FFR	0.7 [0.59, 0.75]	0.74 [0.71, 0.77] */†	0.87 [0.83, 0.92] *	0.87 [0.84, 0.92]	< 0.001
Median CFR	1.5 [1.3, 1.7]	2.5 [2.3, 2.9] *	1.7 [1.6, 1.9] */†	2.5 [2.3, 3]	< 0.001
2D-QCA DS, %	51 ± 11.2	47 ± 8.3 †	40 ± 11.3	45 ± 10.5	0.024
3D-QCA DS, %	61 ± 11.1	53 ± 11.6 †	44 ± 10.3 *	43 ± 13.1	< 0.001
2D-QCA MLD, mm	1.29 [1.09, 1.34]	1.29 [1.03, 1.5] †	1.78 [1.27, 2.08] *	1.55 [1.19, 1.79]	0.003
3D-QCA MLD, mm	0.8 [0.6, 1.1]	1.2 [0.9, 1.5] */†	1.4 [1.2, 1.7] *	1.5 [1.2, 1.8]	< 0.001
2D-QCA Lesion length, mm	9.5 [8.4, 13]	8.7 [7.6, 10.9]	7.3 [6.9, 7.9]	9.7 [6.5, 13]	0.275
3D-QCA Lesion length, mm	17.9 [13.7, 26.7]	23.4 [13.3, 29.6]	13.4 [9.2, 20.6]	16.3 [10.4, 23]	0.025
2D-QCA Area stenosis, %	76.2 [70.9, 76.9]	72.8 [65.6, 78.4]	67.4 [50.6, 74]	71.3 [58.6, 79]	0.067
3D-QCA Area stenosis, %	82 [70.8, 85.3]	73.3 [60.6, 81.2] †	62.9 [53, 71.3] *	61.9 [50.3, 72.3]	< 0.001
2D-QCA Mean RLD, mm	2.71 [2.5, 3]	2.5 [2, 2.8]	2.9 [2.2, 3.4]	2.7 [2.5, 3]	0.168
3D-QCA Mean RLD, mm	2.3 [2.1, 2.6]	2.5 [2.3, 2.7]	2.7 [2.3, 3]	2.7 [2.4, 3.2]	0.007
LAD lesion, n (%)	48 (68%)	60 (74%) *	35 (59%)	140 (65%)	0.281
Normal to mildly reduced CFC, n (%)	16 (23%)	77 (95%) *	27 (44%) */†	202 (94%)	< 0.001
Moderately to severely reduced CFC, n (%)	55 (77%)	4 (5%) *	34 (56%)	15 (6%)	< 0.001

Variables are expressed as n (%), mean ± SD or median (Q1, Q3), where appropriate. * *p*-value < 0.05 compared with FFR+/CFR+. † *p*-value < 0.05 compared with FFR−/CFR−.

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
