# Peer review of "Three-Dimensional Angiographic Characteristics versus Functional Stenosis Severity in Fractional and Coronary Flow Reserve Discordance: A DEFINE FLOW Sub Study"

_diagnostics, 2022, doi:10.3390/diagnostics12071770_

Round 1

Reviewer 1 Report

This is an important study about coronary artery disease (CAD) assessment.  For recent years the debate between morphological approach versus physiological approaches  for  CAD  evaluation results in deeper understanding of the pathology and  necessity of interventions, especially in patients with stable coronary syndromes.  The results of the present study found the importance of physiological analysis (FFR and CFR) even if 3-D stenosis measurement were used. The study is well done and presented and recommended for publishing

Author Response

Reviewer 1

This is an important study about coronary artery disease (CAD) assessment.  For recent years the debate between morphological approach versus physiological approaches  for  CAD  evaluation results in deeper understanding of the pathology and  necessity of interventions, especially in patients with stable coronary syndromes.  The results of the present study found the importance of physiological analysis (FFR and CFR) even if 3-D stenosis measurement were used. The study is well done and presented and recommended for publishing

Response: We thank the reviewer for the efforts put into the reviewing process of our work and the kind words regarding our manuscript.

Reviewer 2 Report

The authors investigated the association 23 between 2D and 3D qualitative coronary angiography (QCA) and each of Fractional Flow Reserve (FFR), 25 Coronary Flow Reserve (CFR) and Coronary Flow Capacity (CFC) to identify lesions asociated with myocardial ischemia. The idea is good, and the methods are well applied. However, I feel that the manuscript needs a serouis editing/style change to be in a publication form.  

- The clinical importance of the findings must be discussed in the discussion (from the Diagnostics point of view).

- The paper needs to be rewrittent/edited and to have an easier flow.

- Many sentences have no references in the introduction, such as the following:

Despite the well-documented mismatch between visual angio-48 graphic stenosis appearance and FFR values, two aspects deserve further evaluation. First, 49 contemporary quantitative angiography analysis (QCA) allows 3-dimensional vessel re-50 construction to provide more detail regarding anatomical stenosis severity, which may 51 provide a better estimation of functional stenosis severity. Second, FFR is known to be 52 influenced by microvascular resistance, where, for a given stenosis, FFR increases with 53 increasing microvascular resistance. Therefore, a comparison with FFR alone may lead to 54 erroneous conclusions regarding the ability of angiography to define functional stenosis 55 severity. The combined assessment of coronary pressure and flow allows to calculate both 56 FFR and coronary flow reserve (CFR), which together provide more insight into the func-57 tional relevance of coronary artery disease. Moreover, these measurements allow the cal-58 culation of stenosis resistance indices and coronary flow capacity as robust markers of 59 functional stenosis severity and vessel flow characteristics, respectively.

- Please fix the following:

(8,9). => (8, 9).

Conclusions Non => Conclusions: Non   **Methods: 

Coronary; physiology;QCA;Fractional Flow Reserve;Coronary Flow Reserve => Coronary; physiology; QCA; Fractional Flow Reserve; Coronary Flow Reserve

stenosis. (1-5). => stenosis (1-5).    ***you may consider tusing [] instead of () 

table 1. => Table 1.

(table 2). => (Table 2).

- Is the following statement accurate?

Author Contributions: JJP received significant institutional research support from Philips Volcano 293 Corporation for this study. TPH and JJP report consultancy fees for Philips-Volcano. MS received 294 institutional research support from the University of Texas Health Science Center at Houston (for 295 the DEFINE-FLOW study). VES, MCM: declaration of interest: none.

- Please remove or adjust the following section:

Funding: Please add: “This research received no external funding” or “This research was funded by 297 NAME OF FUNDER, grant number XXX” and “The APC was funded by XXX”. Check carefully that 298 the details given are accurate and use the standard spelling of funding agency names at 299 https://search.crossref.org/funding. Any errors may affect your future funding.

Author Response

The authors investigated the association 23 between 2D and 3D qualitative coronary angiography (QCA) and each of Fractional Flow Reserve (FFR), 25 Coronary Flow Reserve (CFR) and Coronary Flow Capacity (CFC) to identify lesions asociated with myocardial ischemia. The idea is good, and the methods are well applied. However, I feel that the manuscript needs a serouis editing/style change to be in a publication form.  

We thank the reviewer for the efforts put into reviewing our manuscript and the kind words. We respectfully addressed al of the concerns and adjusted the manuscript accordingly.

  1. The clinical importance of the findings must be discussed in the discussion (from the Diagnostics point of view).

We thank the reviewer for his critical comments and we have adjusted the clinical implications section within the discussion (page 10) to clarify the importance of the findings of the current manuscript regarding the angiographic and functional mismatch.

  1. The paper needs to be rewrittent/edited and to have an easier flow.

We thank the reviewer for this comment. We edited the manuscript based on the critical evaluation of the reviewers and have rewritten parts of the result and discussion section to accommodate to the readability of the paper. (Page 5/7/10)

  1. Many sentences have no references in the introduction, such as the following:

Despite the well-documented mismatch between visual angio-48 graphic stenosis appearance and FFR values, two aspects deserve further evaluation. First, 49 contemporary quantitative angiography analysis (QCA) allows 3-dimensional vessel re-50 construction to provide more detail regarding anatomical stenosis severity, which may 51 provide a better estimation of functional stenosis severity. Second, FFR is known to be 52 influenced by microvascular resistance, where, for a given stenosis, FFR increases with 53 increasing microvascular resistance. Therefore, a comparison with FFR alone may lead to 54 erroneous conclusions regarding the ability of angiography to define functional stenosis 55 severity. The combined assessment of coronary pressure and flow allows to calculate both 56 FFR and coronary flow reserve (CFR), which together provide more insight into the func-57 tional relevance of coronary artery disease. Moreover, these measurements allow the cal-58 culation of stenosis resistance indices and coronary flow capacity as robust markers of 59 functional stenosis severity and vessel flow characteristics, respectively.

We thank the reviewer for this comment and agree that many of the statements made lack the proper referencing. We added the references accordingly throughout the manuscript.

  1. Please fix the following:

(8,9). => (8, 9).

Conclusions Non => Conclusions: Non   **Methods: 

Coronary; physiology;QCA;Fractional Flow Reserve;Coronary Flow Reserve => Coronary; physiology; QCA; Fractional Flow Reserve; Coronary Flow Reserve

stenosis. (1-5). => stenosis (1-5).    ***you may consider tusing [] instead of () 

table 1. => Table 1.

(table 2). => (Table 2).

We thank the reviewer for the thorough assessment of our paper in order to improve the readability. We adjusted the manuscript accordingly.

  1. Is the following statement accurate?

Author Contributions: JJP received significant institutional research support from Philips Volcano 293 Corporation for this study. TPH and JJP report consultancy fees for Philips-Volcano. MS received 294 institutional research support from the University of Texas Health Science Center at Houston (for 295 the DEFINE-FLOW study). VES, MCM: declaration of interest: none.

We thank the reviewer for this comment. We replaced the conflict of interest statements to the just section and added the author contributions accordingly.

  1. Please remove or adjust the following section:

Funding: Please add: “This research received no external funding” or “This research was funded by 297 NAME OF FUNDER, grant number XXX” and “The APC was funded by XXX”. Check carefully that 298 the details given are accurate and use the standard spelling of funding agency names at 299 https://search.crossref.org/funding. Any errors may affect your future funding.

We adjusted the section accordingly: DEFINE-FLOW is an investigator-initiated trial funded by an unrestricted grant from Philips Volcano with additional funding from the Weatherhead PET Imaging Center at the University of Texas Health Science Center at Houston.

Reviewer 3 Report

The current study compared three-dimensional angiographic characteristics versus functional stenosis severity. The study is interesting but several aspects should be improved. Exact definitions should be provided.

1. Statement in lines 52-53. Fractional flow reserve is defined as maximum myocardial blood flow in the presence of a stenosis divided by the theoretical maximum flow in the absence of the stenosis. Based on definition when flow is expressed as pressure gradient divided by resistance, Rmin is resistance at maximum coronary hyperaemia is cancelled out in the ratio. FFR is close to Pd/Pa under conditions of maximum hyperaemic stimulus. What is the foundation for the statement in lines 52-53?

2. Line 102. Coronary flow velocity reserve is a reflection of the coronary flow reserve but is not the same. Absolute coronary flow reserve requires knowledge of the area of the coronary vessel. 3. The authors should better define what they mean by FFR+, FFR-, CFR+, CFR-. The plus and minus are rather misleading or uninformative. 4. Certain statements are without reference e.g. lines 274-276. There appears to be a paucity of references. All specific statements containing scientific knowledge should be backed up by references. 5. Lines 297-300. This has not been completed. The authors should take care that their manuscript is complete. 6. Lines 293-297. Author contributions are not specified. 7. Reference 31 is incomplete. 8. The superior value of any diagnostic approach requires a randomized clinical trial that demonstrates an effect on a clinical outcome.

Author Response

The current study compared three-dimensional angiographic characteristics versus functional stenosis severity. The study is interesting but several aspects should be improved. Exact definitions should be provided.

  1. Statement in lines 52-53. Fractional flow reserve is defined as maximum myocardial blood flow in the presence of a stenosis divided by the theoretical maximum flow in the absence of the stenosis. Based on definition when flow is expressed as pressure gradient divided by resistance, Rmin is resistance at maximum coronary hyperaemia is cancelled out in the ratio. FFR is close to Pd/Pa under conditions of maximum hyperaemic stimulus. What is the foundation for the statement in lines 52-53?

We thank the reviewer for this question. The statement that the FFR is influenced by the microvascular resistance stands from different studies: Meuwissen et al. (Circulation, 2001, PMID: 11208673) and Verhoeff et al. (Am J Physiol Heart Circ Physiol 2012, PMID: 22730389). Variability in microvascular resistance is an important factor in the origin of discordance between FFR and CFR. If a stenosis is present, variability in hyperemic microvascular resistance affects both FFR and CFR in opposite directions. If microvascular resistance is increased, CFR will decrease whereas FFR will increase. Contrary, if microvascular resistance is decreased, CFR will increase whereas FFR will decrease.

We have added the references to the introduction section.

  1. Line 102. Coronary flow velocity reserve is a reflection of the coronary flow reserve but is not the same. Absolute coronary flow reserve requires knowledge of the area of the coronary vessel.

We agree with the reviewer that absolute flow reserve measurements require assessment of the coronary vessel diameters. In order to minimize the effect of vasomotion on the measurements, patients received nitroglycerin into the coronary arteries before quantitative coronary angiography and pressure and flow measurements. In order to avoid misunderstanding, we have adjusted the abbreviation throughout the manuscript: it is CFR for this article indicates coronary flow velocity reserve.

  1. The authors should better define what they mean by FFR+, FFR-, CFR+, CFR-. The plus and minus are rather misleading or uninformative.

We have added a short table with abbreviations and the meaning of the different abbreviations, please see table 1 (As we added a table, table 1 and 2 are now refered to as table 2 and 3, respectively) .

FFR+: FFR≤0.80

FFR-: FFR>0.80

CFR+: CFR<2.0

CFR-: CFR≥2.0

  1. Certain statements are without reference e.g. lines 274-276. There appears to be a paucity of references. All specific statements containing scientific knowledge should be backed up by references.

We have added the references accordingly.

  1. Lines 297-300. This has not been completed. The authors should take care that their manuscript is complete.

We apologize and thank the reviewer for their thorough feedback on our manuscript. We completed the missing sections.

  1. Lines 293-297. Author contributions are not specified.

We added the author contributions.

  1. Reference 31 is incomplete.

We thank the reviewer for this omission at our end. We have completed the reference.

  1. The superior value of any diagnostic approach requires a randomized clinical trial that demonstrates an effect on a clinical outcome.

Thank you for this valuable suggestion, we have added the following sentence to the manuscript in the limitation section:

“Randomized clinical trials are needed to confirm the findings of the present study.” (P. 10, limitation 

Round 2

Reviewer 2 Report

The authors have addressed my major/minor concerns.

Reviewer 3 Report

The authors have provided short but acceptable answers to the comments of the reviewer.